# Criterion Validity of Functional Movement Screen as a Predictor of Sports Injury Risk in Chinese Police Staff

**DOI:** 10.3390/ijerph19126992

**Published:** 2022-06-07

**Authors:** Xuejuan Huang, Hua Liu

**Affiliations:** 1Department of Sports Engineering and Information Technology, Wuhan Sports University, Wuhan 430079, China; hxuejuan@whsu.edu.cn; 2Department of Health Science, Wuhan Sports University, Wuhan 430079, China

**Keywords:** criterion validity, police staff, functional movement screen, sports injury risk

## Abstract

The occurrence of sports injury has been proven to be highly associated with injury history. (1) Background: This study aimed to validate the feasibility and effectiveness of the Functional Movement Screen (FMS) as a predictor of sports injury risk for Chinese police staff by exploring the optimal cut-off value of the FMS total score for the identification of previous injury. (2) Methods: More than 160 Chinese police staff were recruited and completed the FMS and interviews. The recorders of 148 (79 females and 69 males) participants met the data analysis requirements. For the goal of evaluating the total score of the FMS as a predictor of sports injury risk, all data underwent statistical analysis, calculation of ROC and AUC, evaluation of threshold validity, and so on. (3) Results: The total score frequency distribution of participants’ FMS presented the shape of a normal distribution. The statistical results of the study showed that the FMS composite scores (10.6 ± 2.28) of the police staff with a sports injury history were indeed less than those without an injury history (12.4 ± 2.26). The threshold of a total score of the FMS test that warned of a sports injury risk in Chinese police staff on the basis of a previous injury history was 13.5, with an acceptable AUC value (0.701). In accord with the real social environment and the rule of natural human physiological change, the FMS results of this study showed a distinct deterioration trend with increasing age. (4) Conclusions: The FMS deserves consideration by trainers and clinicians as a pre-exercise physical examination for Chinese police staff to avoid sports injury.

## 1. Introduction

According to the statistics of the ministry of public security of China, more than 10,000 Chinese police officers have lost their lives before their retirement in the past four decades. The number of police officers who died at a young or middle age hit a record high of 480 individuals in the year 2020. It is deplorable that death due to overwork accounts for more than half of the deaths of Chinese police officers. In sharp contrast to the Chinese situation, there were only 128 American police deaths in 2017, the lowest number in the history of the past five decades. Most of these deaths were caused by traffic accidents or shootings, with only 10 unexpected deaths. Unfortunately, most of the deaths of Chinese police officers counted in the above data happened suddenly, without any warning. Death can occur when a police officer is sitting in the office, walking on a road, lying in a bed, or in other situations. The above realities reveal a health crisis for Chinese police officers and show that the police population experiences a high occurrence of various chronic illnesses and is vulnerable to intense work and high pressure. Exercise is good for human health; this academic concept was the focus of a project involving exercise to promote health put forward jointly by the American College of Sports Medicine and the American Medical Association in November 2007. However, exercise cannot promise health promotion because improper exercise can lead to sports injuries. Wikipedia defines “sports injury” as an injury that occurs during sports, athletic activities, or exercise. The efficiency of improving health through exercise depends not only on the quantity and intensity of exercise load but also on the quality of the movement patterns, which determines the value of physical exercise. The wrong movement patterns do not bring health but instead increase the risk of sports injury. Therefore, one of the important measures to solve the critical health crisis of the Chinese police is to guide them to carry out sports with the correct movement patterns and avoid sports injuries.

Movement patterns are the foundation of the optimum performance pyramid, supporting the balanced adequate development of movement efficiency and sports skills. It is necessary to evaluate movement patterns before physical exercise. The functional movement screen (hereafter FMS) is one of three core tools of functional movement systems, and it should be accomplished before the other two tools. The FMS is a screening method for diagnosing basic human movement pattern disorders, and it can effectively detect the symmetry, stability, flexibility, functional movement quality, and basic movement posture of the human body [1]. The FMS consists of seven evaluation actions: deep squat, hurdle step, inline lunge, shoulder mobility, active straight leg raise, trunk stability push-ups, and rotary stability. It is widely used in international and professional athletes’ sporty ability evaluations to discover an individual’s sports risk factors, assist in the design of training plans, and correct them according to the test results.

The three most important perceived risk factors affecting non-contact injuries are previous injuries, fatigue, and muscle imbalance. The FMS, questionnaires, and isokinetic muscle testing are three screening tests used to detect injury risk [2]. Some research has demonstrated a relationship between FMS total scores and sports injury risk. Those with FMS total scores of less than 14 had a significantly greater likelihood of an injury compared with those with higher scores [3]. At the same time, other research has contradicted the injury predictive value of the FMS, such as an application of machine learning that was conducted to determine adolescents’ optimal cut-off of the FMS total score for predicting future injury by calculating receiver operating characteristic (hereafter ROC) curves, which yielded unacceptable results for the area under the curve (hereafter AUC), sensitivity, specificity, and odds ratio [4].

After comparing these research works supporting the FMS as a predictor of injury, some restrictions on the subjects of the studies were found. The first was a constraint on age. A university-level athlete with an FMS score of less than 17 on the FMS had an approximately 4.7 times greater chance of suffering a lower extremity injury [5]. However, no statistically significant associations were found between total FMS scores and injury status in high school athletes [6]. Second, subjects engaged in different sports had different values of the optimal cut-off of the FMS total score for predicting injury. Amateur male soccer players whose FMS total scores were less than 14.5 had a two times greater injury risk [7]. The same optimal cut-off of the FMS total score was present in the case of Chinese college female basketball players [8]. An FMS score of less than 16 was related to an increased occurrence of injuries in Wushu athletes [9]. Third, the type of work was considered. The research of Clare B. suggested that the FMS could be reliably applied to tactical populations, including police officers, firefighters, and military personnel, as a screening tool and indicated that a score below 14 means an increased risk of injury [10].

Why could a similar type of engaged sports or job lead to the validity of the FMS as a predictor of injury? The reasons exist in real-life activities. Engagement in the same sport or job means similar physical activities, life routines, environments, and social conditions, all of which share a similar possibility of injury occurrence, except for accidents or other forces majeures. On the basis of this discussion, appropriate constraints on the subjects are an essential prerequisite for investigating the validity of the FMS as a predictor of injury risk and exploring the optimal cut-off for the FMS total score.

The occurrence of sports injuries has been proven to be highly associated with an injury history. The purpose of this study was to explore the association between FMS score and sports injury history of different-aged police staff to verify the validity of the FMS as a predictor of injury risk and determine the optimal FMS cut-off score for identifying previously injured individuals in the population of police staff.

## 2. Methods

### 2.1. Subjects

A total of 165 police staff volunteered for the research. They were all recruited from Hebei Province and included 82 males and 83 females, and the range of age was 23–56 years old. A large majority habitually used their right hand (94%) and right leg (84%) as the main bearing limbs. Most participants maintained a certain frequency of physical exercise. Subjects’ experience in the role of police staff ranged from 2 years to more than 30 years.

### 2.2. FMS Protocol

The FMS assessment was performed by 4 qualified physical fitness coaches who had obtained FMS certification at least 2 years previously. Each one was responsible for the judgment of one or two action tests, and each participant’s action was judged by the same administrator to reduce potential inter-scoring discrepancies. The whole procedure of FMS assessment was conducted following the standard FMS^TM^ protocol, and 7 different tests were completed in the following order: deep squat, hurdle step, in-line lunge, shoulder mobility, active straight leg raise, trunk stability push-up, rotary stability test, and three clearing tests. The same standard scoring criteria were performed by all administrators. Every action test was scored in a range from zero to three. The score was zero if physical pain occurred during the completion of a test action. A score of 1 was given if the participant was unable to complete the test action, a score of 2 was given if action compensation occurred during the test, and a score of 3 was given only if the individual completed the test action in a standard mode without pain or compensation.

Furthermore, 3 clearing tests were completed to eliminate painful movement for the shoulder mobility, trunk stability push-up, and rotary stability tests. If pain was reported during the clearing tests, the score for the specific test was 0. Symmetry evaluation is an important part of the FMS assessment test. There are 5 test actions—hurdle step, in-line lunge, shoulder mobility, active straight leg raises, and rotary stability—involved in the bilateral assessments, such as the left and right shoulders or legs. If the left and right scores of these 5 actions were unequal, the lower score was utilized to compute the final score. After the 7 assessment tests were completed and recorded by video, the sum of all scores was calculated. A higher score indicated better performance in movement patterns and a lower risk of sports injury.

An ethical review of human experiments is important and necessary for the protection of human beings. In this study, the human experiment was conducted using the FMS, which does not damage the human body and does not pose potential risks. The ethics of the human experiment were reviewed by the medical ethics committee of Wuhan Sports University, and the IRB number was 2022015.

### 2.3. Research Procedures Design

The research was in the form of a cross-sectional study and was conducted in agreement with the Declaration of Helsinki. The collection of participants’ physical data was carried out through an online questionnaire and face-to-face interviews during the FMS assessment. Some demographic characteristics were recorded, including sex, age, body mass (kg), height (cm), hand and leg dominance, weekly exercise frequency, injury history, and so on. The injury history information was screened according to anatomical locations, including the ankle, shoulder, waist, back, leg, and knee. All included participants were required not to be experiencing an acute attack of disease or injury during the FMS assessment and not to have exercised for at least 30 min before the test.

After the interview and FMS assessment, all collected data were cleaned to ensure the correctness and effectiveness of the subsequent data analysis. In order to remove and correct dirty data, data cleaning included removing meaningless characters (such as extra spaces), deleting duplicate data, checking the consistency of data units, locating and supplementing missing values, handling abnormal values, and so on. The software Microsoft Excel and IBM SPSS for Windows version 26.0 were used for record storage and statistical data analyses. On the basis of the results of the data analysis, the relationship between the FMS total score and injury history was discussed, and some conclusions were drawn.

### 2.4. Statistical Analysis

Statistical analysis was performed with SPSS for Windows, version 26.0. Several statistical indicators were used to express the data, such as the mean, standard deviation (SD), mode, and median or percentage (%) for normally distributed data and the median and mode for skewed data. It was necessary to verify the distribution of the population according to the different total scores on the FMS to confirm that the sampling of participants was reasonable and in line with the actual situation. The physical conditions of recruited police staff were presented through the statistical results of physical index data. The dichotomy method was applied to determine the classification of sports injury possibility. The value of the injury risk of participants who reported a previous injury was set at 1, and others were set at 0.

ROC curves were performed to investigate the optimal cut-offs for FMS total scores to distinguish injured police staff from those without an injury history. The AUC represents the degree or measure of separability of classification presented by the ROC. Only when the value of AUC is greater than 0.7 is the classification supported by the ROC acceptable. The higher the AUC, the better the classification is at distinguishing between police staff with or without high sports injury risk. The ROC curve is plotted with FPR against TPR, where FPR is on the *x*-axis and TPR is on the *y*-axis.

Some other statistics could not be ignored. Whether the difference in sex or exercise habits affects the FMS score needed to be verified. Therefore, a Mann–Whitney test with a 95% confidence interval (CI) was conducted to compare the difference in sex or weekly exercise frequency in the FMS total scores and injury risk, respectively.

## 3. Results

### 3.1. Descriptive Statistics

#### 3.1.1. Physical Condition of Participants

More than 160 police staff participated in the FMS, interview, and information collection, and finally, a total of 148 data records met the consistency and integrity requirements of the data analysis, 79 of which were for females, accounting for slightly more than a half. According to the age division standard of the physical exercise standard test for Chinese police, the participants were divided into four age groups: less than 30, more than 31 and less than 35, more than 36 and less than 40, and more than 41 and less than 56. It should be pointed out that in order to make the number of subjects in each group as close as possible, all people over 40 were placed into one group. As shown in Table 1, the average of the participants was 34, and the median age was also 34. The average heights of the four different age groups were all nearly 175 cm (male) and 163 cm (female), and the participants’ weights increased with age. Consequently, participants’ body mass index (BMI) also increased. The BMI values of more than 60% of participants were within the normal range, and 34.5% of police staff were overweight. A total of 25 police staff reported a sports injury history, with an especially high incidence in police staff over 35 years old. On the basis of the above, it can be concluded that the physical condition of all participants was barely satisfactory.

The exercise habits of the police staff participating in the FMS were described by weekly exercise frequency. The details of the weekly exercise frequency of different age groups are presented in Table 2. Most of these police staff exercised occasionally (35.1%) or a little every week (30.4%). Those who reported a history of sports injury in the past year were represented in all categories of weekly exercise frequencies. Among them, the police staff who exercised occasionally reported the highest previous sports injury rate (36% of all injury reports). All reported injury information was recorded by anatomical location, and the collected results included the ankle, shoulder, waist, back, leg, and knee.

In order to analyze the correlation between sex or exercise habits and the sports injury history of the police staff volunteers, the hypothesis that sex or exercise habits have no direct relationship with the sports injury history was put forward. The companion probabilities calculated under the t-test framework using SPSS software were 0.700 and 0.739, respectively. Because the values of the two results were both greater than 0.05, neither of the above assumptions could be rejected. Therefore, it was concluded that there were no significant differences between sex or weekly exercise frequency with sports injury history in the population of Chinese police staff.

#### 3.1.2. FMS Scores of Participants

All police staff recruited from Hebei Province completed the FMS as volunteers. Their FMS total scores ranged from 6 points to 18 points. The graphical shape of the number distribution of participants with different scores was very similar to that of the normal distribution, as detailed in Figure 1. The top three FMS total scores that clustered with the largest number of participants were in the following order: 13 points, 10 points, and 12 points. Figure 2 shows the details of the frequency of different scores of the seven FMS actions. Higher scores correspond to a higher quality of test action completion. Most participants performed best in the action of the active straight leg raise and worst in the action of the deep squat. The quality of the other actions was mediocre, except for the trunk stability push-up, an action with a polarized completion condition. Table 3 presents the distribution of the total scores and single scores of each test action in the group for all ages and different ages. The average of the total scores of all participants was 12, and the total score showed a decreasing trend with age until reaching a trough at the age range of 36–40, with a slow upward trend after the age of 40. Most participants obtained only 1 point in the completion of the test action of the deep squat, which is consistent with the situation shown in Figure 2. A certain number of participants could not complete the action of the trunk stability push-up or reported pain and only scored zero. All aforementioned statistical results discovered an alarming situation of movement pattern quality in the police staff who completed the FMS.

### 3.2. ROC Curve and AUC Analyses

The ROC curve was first calculated with valid records of all participants to determine an appropriate cut-off score to support an early warning of sports injury risk using the FMS in the population of Chinese police staff. The visualization of calculation results by SPSS is shown in Figure 3. The AUC result of the ROC curve was 0.701, which meant an acceptable validity of prediction judgment based on this calculation for sports injury risk in Chinese police staff. The Youden index (or Youden’s J statistic) was defined to find the optimal cutoff value using the following formula [11]:J_max_ = max_t_{sensitivity(t) + specificity(t) − 1},(1)

Where the maximum Youden index is reported, t denotes the threshold for the binary classification. On the basis of the method mentioned above, the threshold of the total score of FMS for warning of sports injury risk was 13.5, which is slightly lower than the threshold of 14 commonly reported in previous studies. Concerning the fluctuation of participants’ FMS scores with age, the ROC and AUC were calculated separately using the data records of different age groups, and these results are visualized in Figure 4. The AUC values of the four ROC curves for the groups labeled as aged less than 30, aged 31–35, aged 36–40, and aged more than 40 were 0.717, 0.707, 0.675, and 0.789, respectively, which satisfied the validity requirement. Correspondingly, for the four age groups, the thresholds of the optimal cut-off for FMS total scores were 13.5, 12.5, 10.5, and 13.5, respectively.

Injury history has been proven to be highly correlated with future sports injury probability. If the FMS total score can be used to effectively identify the history of sports injury, it can also be used to predict the risk of sports injury. Because sports injury sometimes causes irreversible damage to the body, our research study adopted a stricter test principle of prediction validity to have better specificity and sensitivity. Therefore, an indicator describing how many actual positive cases were effectively identified was calculated to verify the validity of the optimal cut-off FMS total score for injury risk prediction. In the subject population of all ages, on the basis of the threshold of FMS total score (13.5), 96% of police staff with sports injury history were distinguished from all of those with a sports injury history, except for one young male police officer with a history of ankle injuries, an age of 26, a height of 181 cm, a BMI of 22.9, and a score of 15 on the FMS.

Different thresholds were obtained according to the group division based on age and presented a large difference in the performance of prediction validity. The FMS total score thresholds of the groups less than 30 years of age and more than 40 years of age were also 13.5 and obtained 100% recognition of sports history injuries. The optimal cut-off of the FMS total score of the police staff group of more than 31 and less than 35 years of age was 12.5 and detected 83.3% of previous sports injuries. The FMS total score threshold of the group of more than 36 and less than 40 years of age was the lowest, 10.5, and presented a less-than-perfect performance of recognition of previous sports injuries. It missed three of the eight cases with a sports injury history. The study further investigated the cause and found that the number distribution of people with different FMS total scores in that group (as detailed in Figure 5) was the reason: the number of the police staff who obtained 10 points or 13 points had the same two peaks of seven individuals, which led to the threshold value of 10.5 representing the optimal cutoff of FMS total score by a narrow margin.

The average of these four thresholds was 12.5, and the homologous standard deviation was 1.41. All the above descriptions showed that a certain degree of threshold fluctuation occurred when FMS was a predictor of sports injury risk in Chinese police staff of different ages. This is consistent with the actual work intensity and life pressure of different-aged people in China.

## 4. Discussion

The purpose of this study was to verify the validity of the FMS for predicting sports injury risk in the police staff population. As the statistical results showed, the total FMS score frequency distribution of participants recruited from Chinese police staff in Hebei Province was approximately normal. This retrospective research application of the FMS assessment method in the Chinese police staff population is the first to be published and worthy of being accepted as a prospective study for evaluating whether FMS can be used to predict sports injury risk in the police staff community. A cut-off FMS total score of 13.5 provided acceptable specificity and sensitivity for distinguishing previously injured police staff from those without a sports injury history.

The intended purpose of movement screening focuses on providing a systematic tool to monitor movement patterns by creating a functional movement baseline to identify individuals at risk who are maintaining or improving physical exercise [1]. The FMS is comprised of seven fundamental movement activities tests designed to provide observable performance of basic locomotor, manipulative, and stabilizing movements. As the weakness and imbalance of movement patterns are reinforced, individuals are at an increased potential for micro- or macro-traumatic injury [12]. According to the minimum mean FMS scores in healthy young active individuals, an untrained individual with less than the cut-off score of 14 points could be identified as presenting compensation patterns and a high risk of injury [13]. Although it has been suggested that the total FMS score should not be used for predicting injury risk because of individual differences in scores, most questions focus on whether a total score greater than 14 means a lower relative risk [12].

Previous studies have highlighted that lower scores on the FMS are associated with an elevated risk of sports injuries. For example, a significant correlation was found between low-scoring (≤14) athletes and injury [14]. However, some researchers argued that the strength of association between the FMS composite score and subsequent injury does not support its use as an injury prediction tool [15]. Furthermore, a predictive relationship between FMS composite score and injury history and the development of future injury was demonstrated: athletes with an FMS composite score of 14 or less combined with a self-reported injury history were at a 15 times increased risk of injury [16]. The statistical results of this research showed that the FMS score (10.6 ± 2.28) of these police staff with sports injury histories were indeed lower than those without injury histories (12.4 ± 2.26). It was reported that previous injuries affected FMS performance according to the screening tests of 200 American national collegiate Division I athletes [17]. This indicates that the FMS score could be a potential predictor of sports injury in the population of Chinese police staff on the basis of previous injuries. The true positive rate of the threshold (13.5 points) of the FMS total score as a predictor of sports injury in Chinese police staff was 96% when combined with the injury history.

The physical exercise standard test for Chinese police provides distinct evaluation benchmarks for different age groups. It was consistent with the FMS score fluctuation with the police staff’s age in this research. This phenomenon is supported by some realistic social facts: the older the age and the higher the position, the greater the responsibility and the greater the pressure of supporting a whole family. On the other hand, the performance of the physiological function of the body begins to decline after the age of 30, which is not a negligible factor. Therefore, this study explored the distinct thresholds of the FMS total score for predicting sports injury risk according to the age divisions of the physical exercise standard test for Chinese police staff. Unsurprisingly, the optimal cut-off score for the FMS hit the lowest point in the age range of 36 to 40 years. The research work verified the necessity of setting different thresholds according to age when using FMS as a predictor of sports injury in the population of Chinese police staff.

The threshold for the optimal cutoff of the FMS total score was obtained by comprehensively analyzing actual anthropometric data and FMS results of 148 Chinses police staff in this study. The research results were suitable for a certain range of practical applications. Some other anthropometric data associated with sports performance and the risk of a sports injury, such as body fat, upper and lower limb strength, core strength, and balance, deserve to be added in future studies. Furthermore, the definition of sports injury is broad, so sports-related injury encompasses injuries that occur in a wide range of activities, such as athletic training, fitness exercise, physical activities, and so on. Chinese police staff who spent most of their time at work were the study’s subjects. Their responsibilities were different from those in charge of apprehending offenders. A study conducted in the United States exploring the association between FMS score and a worker’s compensation claim found that the FMS did not predict future occupational injuries among Denver Fire Department firefighters [18]. All injuries caused by sports are categorized as sports injuries, including injuries produced because of work activities during working hours that meet the identification standard, namely, occupational injuries. The association between a more general sports injury risk of Chinese police and FMS score deserves further research in the future.

Last but not least, the research work of Joseph et al. (2017) explained that pain was associated with a slightly higher injury risk than a composite score of ≤14 points [19]. Pain excluding physical disease always means poor quality of movement and a history of injury. Pain and poor quality of movement could be reported during FMS. Additionally, asymmetry or a low single FMS score was found to be a better predictor of musculoskeletal injury in American national collegiate Division II athletes than the total FMS score [20]. For these reasons, it is interesting and necessary to study the association between one or more pain reports during an FMS test and the risk of sports injury.

## 5. Conclusion

In summary, because of the low cost and simplicity of implementation, FMS deserves to be considered by trainers and clinicians as a pre-exercise physical examination for Chinese police staff to avoid sports injury. This study provides a threshold of the total score on the FMS, 13.5 points, as a warning for sports injury risk in Chinese police staff based on previous injury history. It is worth emphasizing that the performance of FMS became worse with age. Therefore, older Chinese police staff need to pay more attention to sports injury prevention. Our future research work will focus on two directions: one is the exploration of the association between reported pain without injury history during the FMS and sports injury risk, and the other is research on how to calculate the total score in a more reasonable and reliable way than direct summation.

## Figures and Tables

**Figure 1 ijerph-19-06992-f001:**
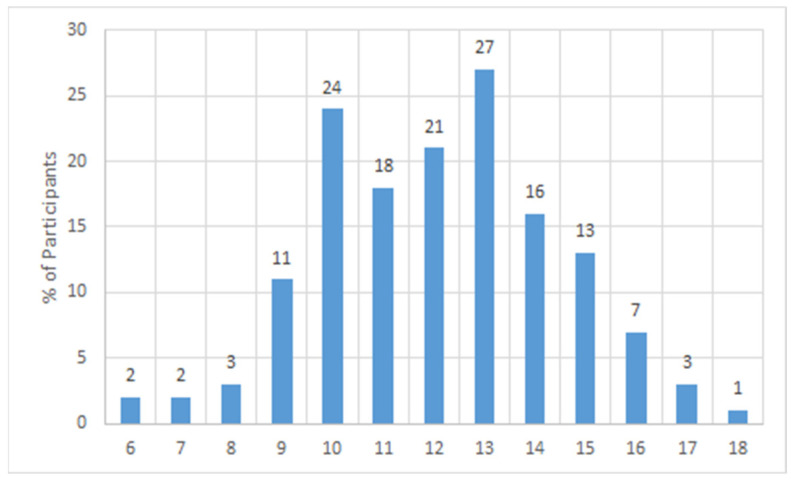
The number distribution of participants with different scores.

**Figure 2 ijerph-19-06992-f002:**
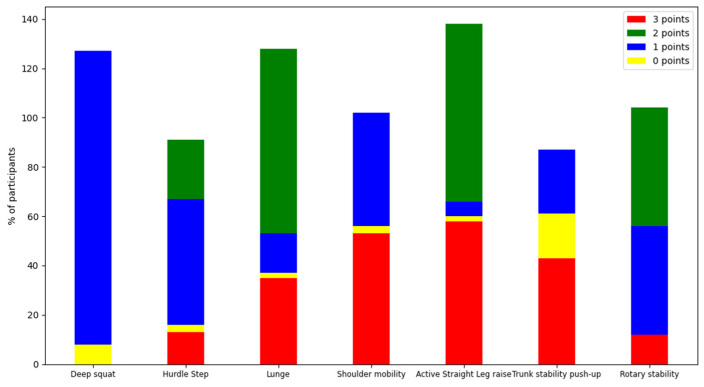
The frequency of different scores of seven FMS actions.

**Figure 3 ijerph-19-06992-f003:**
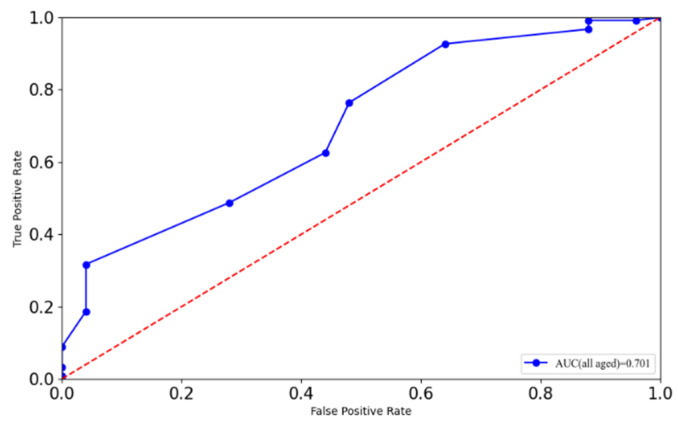
Receiver operation characteristic (ROC) curves for FMS total scores as a predictor for all participating Chinese police staff.

**Figure 4 ijerph-19-06992-f004:**
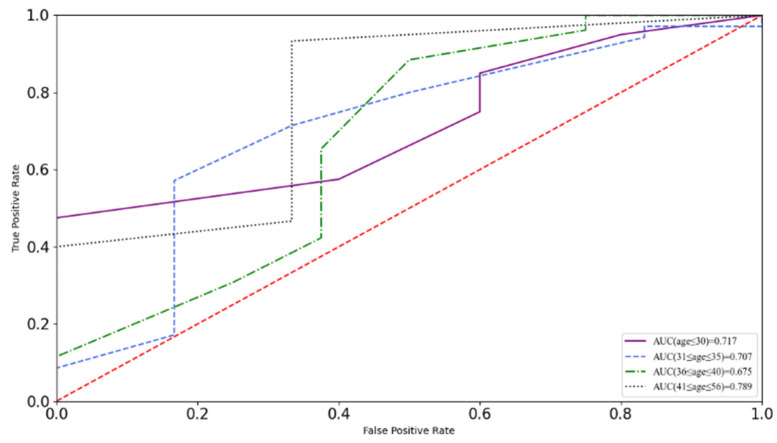
Receiver operation characteristics (ROC) curves for FMS total scores as a predictor for four age groups of Chinese police staff.

**Figure 5 ijerph-19-06992-f005:**
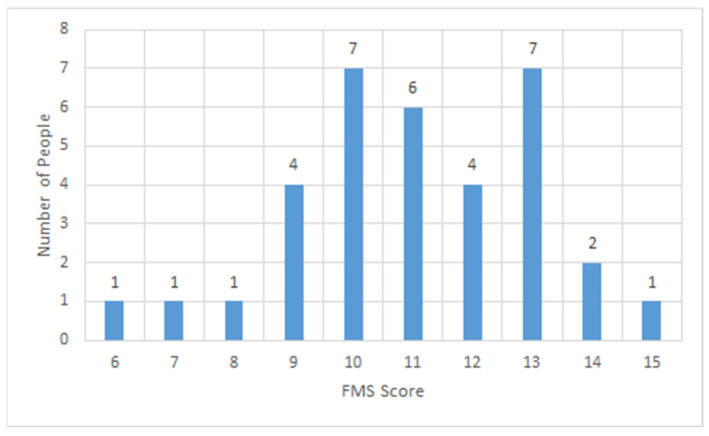
The number distribution of people with different FMS total scores in the group of more than 36 and less than 40 years of age.

**Table 1 ijerph-19-06992-t001:** Demographic characteristics of the subjects. Data are presented as means ± SD, mode, and median or percentage (%).

Range	Sex (Total, Females)	Age Distribution	Height (Male, Female)	Weight (Male, Female)	BMI (Male, Female)	Injure Rate
All	148, 79 (53.4%)	34 ± 7.43, 31, 34	175 ± 4.27, 163 ± 3.77	74 ± 11.42, 58 ± 8.48	24 ± 3.76, 22 ± 3.08	25, 16.9%
Age ≤ 30	45, 24 (53.3%)	25 ± 2.78, 26, 26	176 ± 3.81, 164 ± 3.59	70 ± 10.39, 56 ± 7.39	22 ± 3.44, 21 ± 3.01	5, 14.7%
31 ≤ age ≤ 35	41, 23 (56.1%)	33 ± 1.54, 31, 33	175 ± 4.62, 163 ± 3.67	74 ± 15.35, 58 ± 10.30	24 ± 3.82, 21 ± 3.87	6, 14.63%
36 ≤ age ≤ 40	34, 16 (47.1%)	38 ± 1.08, 39, 38	175 ± 4.65, 164 ± 4.68	75 ± 14.12, 61 ± 7.45	24 ± 4.51, 22 ± 2.20	8, 23.53%
41 ≤ age ≤ 56	28, 16 (57.1%)	45 ± 4.45, 41, 42	174 ± 4.00, 163 ± 3.34	76 ± 8.23, 61 ± 7.10	25 ± 2.41, 22 ± 2.49	6, 21.43%

**Table 2 ijerph-19-06992-t002:** Distribution of Chinese police staff with different weekly exercise frequencies. Data are presented as percentages.

Weekly Exercise Frequency	All Ages	Age ≤ 30	31 ≤ Age ≤ 35	36 ≤ Age ≤ 40	41 ≤ Age ≤ 56	Injure Rate
Never	8.8%	4.4%	12.2%	11.8%	7.2%	16%
Occasionally	35.1%	33.3%	34.1%	38.2%	35.7%	36%
1–2 times per week	30.4%	42.2%	24.4%	26.5%	25.0%	20%
3–4 times per week	15.6%	17.9%	17.1%	14.7%	10.7%	20%
≥5 times per week	10.1%	2.2%	12.2%	8.8%	21.4%	8%

**Table 3 ijerph-19-06992-t003:** Total and single-action scores for all ages and different-aged subjects. Data presented as means ± SD.

Test Action	All Ages	Age ≤ 30	31 ≤ Age ≤ 35	36 ≤ Age ≤ 40	41 ≤ Age ≤ 56
Deep squat	1 ± 0.38	1 ± 0.40	1 ± 0.32	1 ± 0.35	1 ± 0.43
Hurdle step	2 ± 0.66	2 ± 0.67	2 ± 0.64	1 ± 0.60	2 ± 0.62
Lunge	2 ± 0.64	2 ± 0.63	2 ± 0.65	2 ± 0.70	2 ± 0.54
Shoulder mobility	2 ± 0.88	2 ± 0.80	2 ± 0.85	2 ± 0.86	2 ± 0.82
Active straight leg raise	2 ± 0.64	2 ± 0.69	2 ± 0.64	2 ± 0.58	3 ± 0.58
Trunk stability push-up	2 ± 1.01	2 ± 0.87	2 ± 1.03	2 ± 1.01	2 ± 1.17
Rotary stability	2 ± 0.58	2 ± 0.50	2 ± 0.51	2 ± 0.53	2 ± 0.70
Total score	12 ± 2.35	13 ± 2.22	12 ± 2.39	11 ± 2.05	12 ± 2.43

## Data Availability

The datasets generated during and/or analyzed during the current study are not publicly available but are available from the corresponding author, who was an organizer of the study.

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
