# Peer review of "Criterion Validity of Functional Movement Screen as a Predictor of Sports Injury Risk in Chinese Police Staff"

_ijerph, 2022, doi:10.3390/ijerph19126992_

Round 1

Reviewer 1 Report

Dear Authors

the paper addressed a relevant topic and cousl be clinically relevant.

The only question is regarding basic factors che could influence questionaire, such as hand grip across population.

An other question is related to the sexes. Could results showed by gender?

Author Response

Thanks for your precious comments on our manuscript. All opinions are very valuable and can improve the manuscript very well. According to these opinions, the manuscript was revised carefully.

Reviewer 2 Report

Xuejuan Huang and Hua Liu report criterion validity of FMS as a predictor of sport injury risk in Chinese police staff. This study is potentially interesting. However, there are several concerns listed below. This paper will be strengthened by addressing the following issues.

1. This study is for human, therefore authors should include IRB protocol number. 

2. FMS is a tool for sports injury prediction. This study is for Chinese police staff. So, they should use workplace injury (not sports injury).

for ex) In discussion part, line 322,

"The research work verified the necessity of setting different thresholds according to age for supporting FMS as a predictor of sports injury  in the population of Chinese police staff "

They should check sports injury of this manuscript  in general. 

3. In recent study (PMID: 32995055) they examined the ability of FMS to predict occupational injury among Denver Fire Department firefighters.

According to this study, although the FMS may be a useful measure of fitness amongst firefighters, it did not provide information on which firefighters may be more at risk to sustain a workplace injury.

Therefore, Xuejuan Huang and Hua Liu should refer this paper  (PMID: 32995055) and discuss it. 

Author Response

(The authors gave the same response as above.)

Reviewer 3 Report

This study presents a new investigation on the prediction of sport injury risk by Functional Movement Screen in a sample of Chinese police. The purposes of the study were well defined and the methods were adequately described. More generally, the topic covered is of scientific interest, the article is well written, and the statistical analysis seems correct.

I suggest that the authors strengthen the discussion by reporting the strengths and weaknesses of the study. In particular, the limitations should include the use of anthropometric data reported by participants.

 Finally, here are my last minor remarks:

-line 97: indicate the percentages of use of right hand and leg.

-line 110: delete the pleonastic expression “, respectively 0,1,2,3”.

- line 129 and following: this is not a study in social work, thus the term gender should be replaced with sex.

- lines 173-175: Explain why all the subjects were divided into five-year age groups except for the last group.

- lines 175-176 and Table 1: height and weight differ significantly between sexes in humans. It is necessary to present these data separately by sex.

-line 269: replace aged with age.

-line 327:  replace “Joseph A. A., etc.,” with “Joseph et al. (2017)”.

-line 330: replace “bad quality of movement quality” with “bad quality of movement”.

- lines 335-336: unclear sentence: revise English.

Author Response

(The authors gave the same response as above.)

Round 2

Reviewer 2 Report

 I support this work is suitable for publication in IJERPH.